# WetSpa-Urban: An Adapted Version of WetSpa-Python, A Suitable Tool for Detailed Runoff Calculation in Urban Areas

**Nahad Rezazadeh Helmi** *, **Boud Verbeiren**, **Charlotte Wirion**, **Ann van Griensven**, **Imeshi Weerasinghe** and **Willy Bauwens**

Department of Hydrology and Hydraulic Engineering, Vrije Universiteit Brussel (VUB), 1050 Brussels, Belgium; Boud.Verbeiren@vub.be (B.V.); Charlotte.Wirion@vub.be (C.W.); Ann.Van.Griensven@vub.be (A.v.G.); Imeshi.Nadishka.Weerasinghe@vub.be (I.W.); wvc.bauwens@gmail.com (W.B.)

* Correspondence: nahad.rezazadeh.helmi@vub.be; Tel.: +32-492-87-1549

**Abstract:** A tool called WetSpa-Urban was developed to respond to the need for precise runoff estimations in an increasingly urbanized world. WetSpa-Urban links the catchment model WetSpa-Python to the urban drainage model Storm Water Management Model (SWMM). WetSpa-Python is an open-source, fully distributed, process-based model that accurately represents surface hydrological processes but does not simulate hydraulic structures. SWMM is a well-known open-source hydrodynamic tool that calculates pipe flow processes in an accurate manner while runoff is calculated conceptually. Merging these tools along with certain modifications, such as improving the efficiency of surface runoff calculation and simulating flow at the sub-catchment level, makes WetSpa-Urban suitable for event-based and continuous rainfall–runoff modeling for urban areas. WetSpa-Urban was applied to the Watermaelbeek catchment in Brussels, Belgium, which recently experienced rapid urbanization. The model efficiency was evaluated using different statistical methods, such as Nash–Sutcliffe efficiency and model bias. In addition, a statistical investigation, independent of time, was performed by applying the box-cox transformation to the observed and simulated values of the flow peaks. By speeding up the simulation of the hydrological processes, the performance of the surface runoff calculation increased by almost 130%. The evaluation of the simulated 10 minute flow versus the observed flow at the outlet of the catchment for 2015 reached a Nash–Sutcliffe efficiency of 0.86 and a bias equal to 0.06.

**Keywords:** GIS; hydrodynamic; rainfall–runoff; software; urbanization

## 1. Introduction

The most recent study done by the UN Population Division in 2018 [1] estimated that more than 55% of the world's population was living in urban areas. Rapid growth and development in urban settlements will continue, with almost 60% of people expected to live in urban areas by 2030 [1]. Rapid urbanization has a significant impact on urban runoff, with the natural environment being replaced by impervious materials, including concrete and asphalt. Sealing the urban surface increases the heterogeneity and complexity of land and results in changes to infiltration capacity of topsoil due to soil compaction [2] and its geomorphological characteristics, such as slope [3]. Due to these transformations, both the quality and the quantity of stormwater runoff are prone to change. Urban hydrological processes are complex because of the presence of various barriers causing flow diversion, different building storage capacities, complex geometry, etc. [4].

In order to understand the hydrologic behavior of urban areas and tackle urban problems such as flood and combined sewer overflow (CSO), a reliable rainfall–runoff model is needed [5]. Singh

and Woolhiser [6] found that the first generation of rainfall–runoff models goes back to the 19th century. The current versions capable of simulating runoff are based on models developed by US governmental agencies in 1970 [7]. Several urban rainfall–runoff modeling tools have since been developed by different organizations, and the development of new/adapted/improved models will most likely never stop. Salvadore, Bronders, and Batelaan [5] analyzed 43 well-known catchment scale ($\geq$10 km$^2$) or city scale (<10 km$^2$) rainfall–runoff modeling tools considering both hydrology and hydraulic behavior applicable to urban areas, of which only a few are specialized for urban studies. In general, hydrologic/hydraulic urban models are categorized based on their (1) spatial and temporal resolution, (2) flow routing methods and hydraulic concepts, (3) level of detail regarding hydrological processes, and (4) incorporation of geographic information systems (GIS) and remote sensing data [5,8].

With respect to representation of spatial and temporal resolution in rainfall–runoff models in urban areas, two distinct trends are observed: (1) A full spectrum of spatial (grid based: 10 m to 10 km and non-grid based) and temporal (ranging from minutes to daily to larger timesteps) resolutions covered by catchment scale tools, and (2) tools that are applicable at the city scale, which follow completely different trends. Spatial resolutions are categorized into two discrete groups: Very high-resolution (<10 m), such as semi-urbanized runoff flow (SURF) [9] and the hyperbolic model [10], and non-grid based, for instance, the Model for Urban Stormwater Improvement Conceptualization (MUSIC) [11,12] and Kanal-Regenentlastung (KAREN) [12,13]. The time-steps are similar for both groups when considering temporal resolutions, and range from seconds to days [5]. This shows a gap in urban hydrological models to cover coarser spatial resolutions, i.e., >10 m, which are essential in areas with limited high-resolution remote sensing and GIS data. In addition, due to high heterogeneity in urban areas and fast dynamics of rainfall–runoff responses, there is a demand for a high spatial and temporal resolution hydrologic/hydraulic continuous modeling tool in urban studies [5,8].

In urban areas, due to the presence of hydraulic structures, it is crucial for any modeling tool to be able to simulate these structures in addition to the calculation of overland flow, river routing, and stormwater drainage. The method of calculation can be further categorized into three main groups: (1) Conceptual, (2) hydrodynamic routing methods, and (3) geomorphologic instantaneous unit hydrographs (GIUH or IUH) [5,8]. Due to the short response time of the urban catchment and flash floods, scientists tend to use high-resolution products such as the Model for Urban Sewers (MOUSE) [14], the Storm Water Management Model (SWMM) [15], and InfoWORKS CS [16]. However, surprisingly, Salvadore, Bronders, and Batelaan [5] found that only 15% of the hydrological models in urban areas consider stormwater drainage and sewer systems or other hydraulic structures such as reservoirs, weirs, and pumps in their simulations. These models that capture the detailed representations of the urban drainage system only have a lumped representation of the rainfall–runoff processes at the sub-catchment scale. None of these models allow for detailed spatial representation of the rainfall–runoff processes at the surface. On the other hand, some hydrological models calculate overland flow in a more detailed manner, but they do not have the capability of simulating urban features. The Python version of the Water and Energy Transfer between Soil Plants and Atmosphere (WetSpa-Python) modeling tool [17–19] is a good example of this kind of model, where the diffusive wave approximation method is used for routing overland and channel flow to the outlet without considering any hydraulic structures. Although all of the above-mentioned software packages are suitable and well-known flow simulation tools in urban areas, only SWMM and WetSpa are open-source and freely available [20]. In order to benefit from a large user community, free availability of software is crucial. This need has led to the recent tendency toward using freely available, distributed hydrological models with the capability to simulate typical urban features.

Additional to the increased sealed surface resulting from urbanization, anthropogenic changes in the natural drainage path have a dramatic impact on urban hydrology [8]. The changes in the natural drainage path are mainly caused by manmade separate and combined sewer systems constructed for draining stormwater from impervious surfaces. In general, urban hydrological processes are divided into (1) evapotranspiration (ET), (2) runoff processes, (3) stormwater drainage systems, and

(4) infiltration and subsurface processes, which can be further divided into more sub-groups [5,8]. The capability of urban hydrological/hydraulic models to simulate urban hydrological processes using a more detailed method is essential to better understand the behavior, which can lead to a better estimation of flooding extent and magnitude of CSO events. As an example, evapo(transpi)ration is a crucial element of urban hydrology compared to natural catchments due to the high coverage of impervious surface and dispersed urban vegetation cover [5]. However, the level of detail in the calculation of hydrological processes is different in WetSpa and SWMM. Although both consider evapo(transpi)ration in their flow and water balance calculations, WetSpa calculates these processes in a more detailed, physically-based manner. Additional to evaporation from water in soil surface together with depression storage and intercepted water, WetSpa calculates the transpiration from vegetation for each pixel by using the measured potential of evapotranspiration [21]. Compared to WetSpa, SWMM is less complex in terms of ET calculations, since the calculations only depend on temperature, without the influence of land-use and vegetation [15].

In today's world, due to improvements in computer science and land-mapping, some hydrological modeling tools allow the use of GIS and remote sensing data. Salvadore, Bronders, and Batelaan [5] found that estimating model inputs and parameters and representing the (sub-)catchment surface, including catchment delineation, are the two main reasons to incorporate GIS platforms and remote sensing data in an urban hydrological model.

Recently, some specific studies regarding the use of GIS, remote sensing, and geographical data in urban hydrological modeling with a focus on (sub-)catchment delineation were carried out by researchers [22–25]. This topic can be of interest for any urban catchment, because this superimposition of the surface and sewer topography is a specificity of urban catchments compared to rural ones. Moreover, in (semi-)distributed hydrological modeling, the segmentation of catchments is a real issue. Some approaches, such as the D8 algorithm [26], use only grid-based data to represent surface components for (sub-)catchment delineation [27–29], which are good for natural watersheds but fail when considering manmade underground drainage networks (e.g., sewer systems) in their calculations. To adapt for urban catchments, additional to grid-based data, vector-based data capturing information regarding urban features and hydraulic infrastructures should be considered in this process [22]. The Penn state Integrated Hydrologic Model (PIHM) with GIS interface, called the PIHMgis tool [23], is a good example of this kind of model, which combines catchment delineation processes through the D8 algorithm by use of a digital elevation model (DEM) and a vector-processing step to consolidate multiple hydrologic and hydraulic features, such as stream networks, administrative boundaries, land uses, and drainage networks into one layer in order to calculate the triangular mesh elements. The most recent study, performed by Sanzana et al. [24], used a different methodology by dividing urban areas into two distinct groups of urban features and natural zones. Each of these groups was further represented in the form of urban hydrological elements (UHEs) [30] and hydrological response units (HRUs) [31], respectively. Despite the recent advances in (sub-)catchment delineation by using high-resolution terrain data, the calculation of (sub-)catchment boundaries in urban areas and finding the correct flow path in an accurate manner is still an open scientific discussion [24].

High-resolution space(air)borne imagery products, such as imperviousness cover, land-use, and soil maps together with a DEM are used to calculate some important hydrological parameters [5]. As an example, spatially distributed grid-based information regarding the leaf area index (LAI), the runoff coefficient and roughness are GIS and remote sensing products used by many urban hydrological modeling tools in their fully distributed surface runoff calculation (e.g., WetSpa [19], SURF [9,32], and the GIS-based Urban Flood Inundation Model (GUFIM) [33]). The importance of using remote sensing and GIS data becomes prominent when simulating flow in urban lands by use of distributed parameter maps (slope, runoff coefficient, Manning roughness, etc.) at the pixel level instead of the sub-catchment level. This increase in detail allows additional implementation, for example, of different practices of source-control measures, such as low impact development (LID) [20].

Due to the sheer number of stormwater flow software programs available for urban catchment modeling, the strength of available software tools should be combined. The primary objective of this research was to adapt WetSpa-Python for rainfall–runoff simulation specialized in urban areas. In addition to its high predictive power and calculation of hydrological processes in a fully distributed, physically-based manner, the incorporation of high-resolution GIS and remote sensing data together with high-temporal resolution data made it suitable for this study. As WetSpa-Python was not capable of modeling hydraulic structures, WetSpa-Urban was developed by coupling the surface runoff section of WetSpa-Python [17] together with the underground sewer processes simulated by the high-resolution hydrodynamic model SWMM [15]. SWMM is one of the most well-known open-source hydrodynamic tools for urban studies and its compatibility for coupling with other modeling tools made it a good choice for this study. More realistic rainfall runoff calculations were accomplished by introducing a new approach for sub-catchment delineation to account for both surface and sub-surface features. Finally, to make WetSpa-Urban applicable for both event-based and continuous modeling, three different methodologies were proposed to speed up the modeling calculation time. Using WetSpa-Urban, this study aimed to overcome the shortcomings of both individual models. The new compelling and functional open-source and free rainfall–runoff software package was generated suitable for urban catchments to be used by a larger user community.

## 2. Materials and Methods

### 2.1. The Case Study

In this research, the applicability of the WetSpa-Urban software was tested on the Watermaelbeek (WMB) catchment, which has an area of 6.13 km$^2$ and is situated in the upper Woluwe catchment in the Brussels capital region. Due to its high urban land density (more than 40%), it was an excellent site to assess whether the newly developed software functions correctly in an urban catchment. The elevation in the WMB sub-catchment gently decreases from the south-west, covered by dense vegetation of the La Cambre forest, to the northeast where the outlet is situated. The altitude ranges from 142 m to 60 m above sea level.

As can be seen from Figure 1 (left-hand side), urban land-cover and forested area are the most dominant land-use classes covering almost 38% and 36% of the total WMB catchment. Loamy sand is the primary soil type (70%), followed by sandy clay (17%) and clay loam (13%). According to the data from the Royal Meteorological Institute at Uccle station, the long-term monthly average temperature ranges from 14.5 °C in summer and 5.0 °C in winter. Moreover, the mean yearly rainfall in Brussels is nearly 853 mm/year [18].

Due to limited access to detail information regarding the sewer system, only the main pipes were modeled in this study. Subsequently, the hydraulic networks contained 110 junctions, from which 32 are inlet nodes (shown with red circles in Figure 1 [right-hand side]), with an average distance of 256 m. The length of the modeled sewer pipes was 125 m, on average, and the depth ranged from 1.3 to 2.65 m with an average Manning roughness coefficient equal to 0.0168 (rough form). An offline reservoir (storm basin) with a capacity of 40,000 m$^3$ was connected to the network by a control weir and a pump.

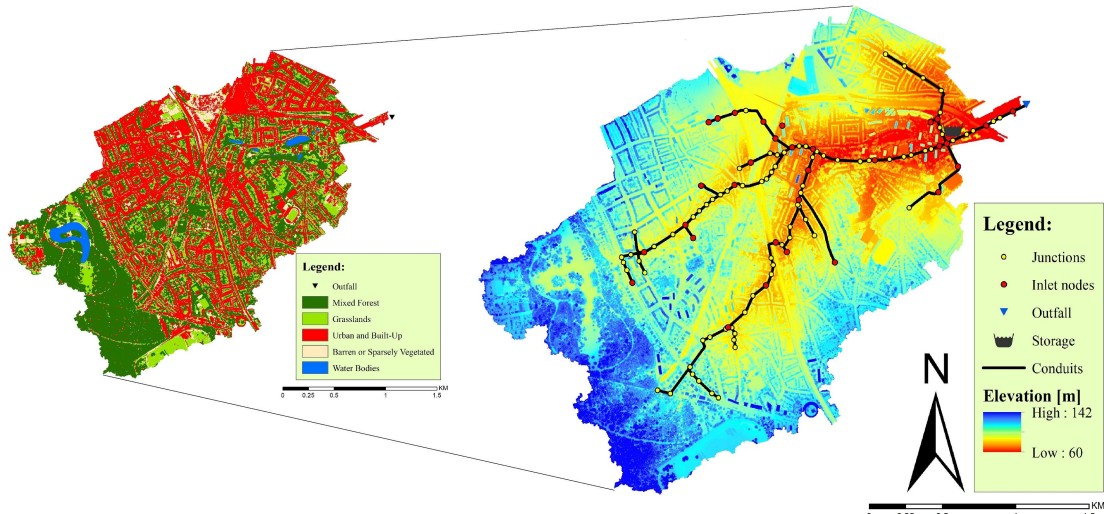

**Figure 1.** High-resolution land-use map [18] and a digital elevation model (DEM) map of the Watermaelbeek (WMB) catchment [34].

### 2.2. The Hydrologic Model: WetSpa-Python

The Python version of WetSpa [17], developed by the Vrije Universiteit Brussel (VUB), is a free-source, GIS-based, fully distributed, physical-based, rainfall–runoff model. Compared to the original WetSpa [35] written in the FORTRAN programming language, the Python version has two main advantages which makes it more suitable for urban studies: (1) The process-based model structure, which allows better understanding of each hydrological process, and (2) the handling of spatial and temporal resolution data, ranging from high to medium. In other words, this version has no limits regarding the simulation time-step and can be varied in the order of minutes to days [17]. Depending on the geomorphological characteristics of each pixel, runoff was calculated and routed to the outlet using the diffusive wave approximation, while the flow discharge at the outlet was calculated through an instantaneous unit hydrograph (IUH) [19].

WetSpa-Python consists of two major components: (1) A GIS-based pre-processing component, which further divides into a fully distributed parameter maps calculator and an IHU processor, and (2) a main physically-based hydrological processes simulator [17]. The model inputs are parameters and meteorological data, spatially distributed over the catchment area. The spatially distributed parameter maps (the full list and their calculation flowchart (Figure A1) are presented in Appendix A) are derived from land-use, soil, and DEM maps together with pre-defined parameter tables and a group of thresholds, but meteorological data (precipitation and potential evapotranspiration time series) are distributed by a thiessen polygon approach in the pre-processing stage. Some of the generated spatially distributed data, such as the travel time of each pixel to the catchment outlet and the standard deviation, the sub-catchment delineation and runoff coefficient maps were subsequently used as an input for the IUH calculator to compute the spatially-distributed flow-response function. Then, the resulting stack of parameters and IUH maps were used in the runoff simulator component as an input to calculate the flow hydrograph at the catchment outlet. The calibration could be done manually by modifying the 8 global parameter sets including: (1) The correction factor for potential evapotranspiration (PET), (2) the actual runoff coefficient correction factor, (3) the rainfall intensity scaling factor, (4) the initial soil moisture correction factor, (5) the interflow scaling factor, (6) the initial groundwater storage parameter, (7) the base flow recession coefficient, and (8) the groundwater storage scaling factor. These global calibration parameters are only used as empirical constants and/or to compensate scaling effects which should be calibrated against observed flow discharge data [17]. This means that in the case of having one measurement station for collecting runoff data at the outlet of the catchment, the 8 global

calibration parameters would be consistent over the whole catchment. In other words, distribution of these global parameters has a direct relationship with the flow observation data.

Due to the heterogeneity and complexity of urban catchments, high-resolution spatial and temporal input data were needed for precise runoff calculations, which required long simulation times, rendering the software inappropriate for long-term simulation. Also, despite its detailed computation of runoff, WetSpa-Python lacked a flow-routing component for the simulation of the flow through hydraulic structures to mimic the discharge in urban studies.

### 2.3. The Hydrodynamic Model: Storm Water Management Model (SWMM)

This software is a well-known, open-source, dynamic, hydrologic–hydraulic (rainfall–runoff) model for the calculation of runoff in urban areas developed by the US Environmental Protection Agency (EPA) [15]. As flash floods and the quick response time of catchments are the main issues in urban areas, the capability of SWMM for both a single event (design purpose) and long-term modeling makes it suitable for urban studies. This can be done with any specified time-step (from minutes to days).

SWMM considers three units for water flow: (1) The atmosphere (evaporation calculation), (2) the terrain-related component, which is further divided into two surface and groundwater components, and finally (3) the conveyance segment (sewer) [36,37]. As SWMM is especially developed for hydraulic modeling of structures such as drainage networks, reservoirs, and weirs, it has high computational power for flow-routing calculations in an urbanized catchment with less focus on distributed hydrological processes. As a result, the surface runoff was calculated based on the non-linear reservoir theory after the subtraction of losses such as depression, infiltration, and evaporation for each sub-catchment separately [15]. Although the sub-catchments were distributed over the study area, the properties of each sub-catchment, such as soil characteristics and slope, were averaged. This made the surface runoff calculation less detailed and conceptual. Conversely, the runoff flow and depth in hydraulic structures (sewer) were simulated in detail using the Barré de Saint Venant equations [15,38].

### 2.4. The WetSpa-Urban Modeling Tool

#### 2.4.1. General Description

WetSpa-Urban is an event-based and continuous rainfall–runoff model for the precise simulation of flow in urban studies at different scales. In other words, it is the coupled version of the two above-described software tools (WetSpa-Python and SWMM), together with some modifications and enhancements for computational efficiency. The strengths of each model were used by coupling the overland and ground processes in WetSpa-Python with the conveyance compartment in SWMM. The general processes modeled in WetSpa-Urban are represented in Figure 2.

In this framework, the flow at the outlet was calculated in two separate sections: (1) The physically-based surface runoff calculation using the WetSpa-Python scripts and (2) the calculation of the flow in hydraulic structures using the SWMM codes. In the first section, all losses (depression, evapotranspiration, interception, etc.) were subtracted from the precipitation and then overland-, inter-, and base-flow at the outlet of each sub-catchment were calculated. In the next step, the overland flow simulated with WetSpa-Python was routed through hydraulic structures (conduits, reservoirs, weirs, etc.) to the catchment outlet using SWMM. In addition to coupling both models, the model structure was adapted to be more user-friendly, applicable for an urban catchment, and efficient for continuous modeling (the link accessing the WetSpa-Urban source codes and pre-installation requirements can be found in the Supplementary Materials section). Three major modifications were adopted: Sub-catchment delineation, improved efficiency of runoff calculation, and a graphical user interface integrating both the WetSpa and SWMM input and running.

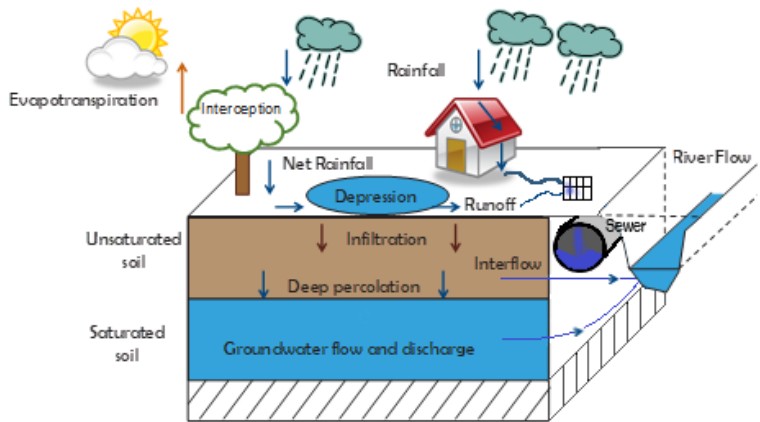

**Figure 2.** General hydrological and hydraulic processes used by WetSpa-Urban—the completed version of the WetSpa-Python general processes [17].

### 2.4.2. New Definition for Sub-Catchment Delineation

The complex geometry of urban landscapes makes it necessary to simulate surface runoff at the sub-catchment scale. As the original WetSpa-Python was developed for the calculation of overland flow at the catchment scale, the model was modified to further simulate at sub-catchment level. Moreover, in urban catchments, due to the presence of manmade drainage networks, the runoff was drained to the outlet of the catchment in two different steps: (1) The above-ground step, in which the water drained to the sub-catchments outlet following the geometry of the terrain, and (2) the underground step, where the water drained by following the slope of the sewer pipes. As a result, the methodology used in the hydrological models has limitations in urban catchments because these models compute flow direction using only DEM information for delineating sub-catchments. To this end, in this study, a new concept was developed where raster and vector processing are the main two steps for (sub-)catchment delineation considering surface and subsurface features. In other words, the sub-catchments were defined by considering both the flow accumulation map generated based on the DEM and the division of drainage zones determined by the position, height, and slope of nodes and pipes. This allowed for the runoff generated by areas following the opposite direction of the sewer pipes to be redirected to the correct manholes, as an example. Table 1 represents information about the data used in this urban catchment delineation method performed during the set-up of the WetSpa-Urban modeling tool for the WMB sub-catchment.

**Table 1.** Information about the data used for sub-catchment delineation in the WMB catchment.

| Source | Provider | Contents | Type |
|---|---|---|---|
| Administrative boundary | Ministry of the Brussels region | Sub-catchment boundaries | Paper |
| 5 m DEM | Urbis-DTM | Elevation of each pixel | Raster |
| Hydraulic structure data | VIVAQUA | Sewer pipe network | Vector |

To this end, as WetSpa-Urban only handles maps in the ASCII format, a sub-catchment boundary map with the same format was used in order to identify the outlet of each sub-catchment. The preparation of the boundary map was done by using a high-resolution DEM map as an input for the D8 algorithm [26]. Other methods, such as the multiple flow direction [39] and the D-infinity method [40], exist and might lead to more accurate results compared to the D8 method. However, as high-resolution data was used in this study, the error induced by the D8 method could be comparable with the uncertainty level of the input data, which could be performed with PCraster-Python GIS functions [41,42], ArcGIS, QGIS, or any other software capable of delineating the sub-catchment boundaries based on the specified location of their outlets (inlet manholes).

The generated sub-catchments were then overlaid with the sub-catchment division map based on the sewer network and compared to administrative boundaries. The usage of administrative boundaries was especially useful in urban catchment with limited access to detailed information regarding the connectivity of each drainage area and their respective manholes (inlet nodes). This procedure further corrected the boundary map that was created using DEM and sewer slope. The other reason for using administrative boundaries in the WetSpa-Urban model was to provide sufficient information, such as runoff estimation, water-balance, and physical parameters for each of the administrative zones, which could be useful for stakeholders, scientists, and water companies. For instance, when implementing LID practices in an administrative zone, the WetSpa-Urban tool assesses its impact on runoff reduction and the extent of flooding within that specific zone. In this method, the aim of sub-catchment delineation was to provide a boundary map that was as close as possible to the extent of the administrative boundaries by taking into account surface slope and drainage network properties.

The comparison between the sizes and location of each sub-catchment showed whether to modify the division or not. It is important to note that the selection of the number of sub-catchments was dependent on the purpose of the study and the availability of input data. In some cases, due to the lack of availability of detailed data regarding drainage networks and/or similarities in characteristics of administrative zones, aggregation into coarser sub-catchments could be achieved. In other cases, due to differences in terrain slope and sewer pipes, the administrative regions should be disaggregated. In water quantity studies, such as evaluating the number of CSO events or flooding, simplifying reality by having fewer sub-catchments does not significantly affect the reliability of the result and improves the computational efficiency. This means a trade-off should be obtained between data availability, the purpose of the study, and simulation efficiency.

### 2.4.3. Speeding up the Surface Runoff Calculation

An additional constraint of the original WetSpa-Python is its slow calculation speed for continuous modeling. This is mainly due to the use of high-resolution remote sensing and GIS input data (e.g., $2 \times 2$ m$^2$). In other words, the input resolution has a direct relationship with the computation time of the modeling.

The second important reason for the low computational speed of WetSpa-Python is due to the lower effective performance of Python compared to other languages such as C, C++, and FORTRAN [41]. To make the new software more user-friendly and applicable to use for long-term simulations, the model was sped-up using three different methods.

First, the programming technique called multi-threading was applied using Python along with the process and system utilities (Psutil) library. The multi-threading approach allowed parallel calculations of surface runoff for a specific number of sub-catchments simultaneously. The Psutil optimized system utilizations such as the computer processing unit (CPU), memory, etc., and the number of sub-catchments that could be run in parallel.

Code optimization was the second approach for speeding up calculation time. As WetSpa-Python is open-source, the codes are accessible, therefore, simplifying the codes and rewriting the equations helped to speed up the model performance. In this way, the equations were modified by defining some constants to avoid repeated calculation, thereby leading to an increase in model performance.

As a third approach, a reduction in calculation time was obtained by imposing a reduction of the maximum length of the instantaneous unit hydrograph (maximum time of concentration = Maxt) without changing the input map (reality). The hydrograph at the outlet of each pixel was calculated using Equation (1) [19].

$$Q_i(t) = \sum_{\tau=0}^{t-\tau} V_i(\tau) U_i(t-\tau) \qquad (1)$$

where $Q_i(t)$ is the flow at the outlet of each sub-catchment generated from input cell $i$ (m$^3$/s), $U_i(t-\tau)$ is the flow path response function or IUH (L/s), $\tau$ is lag time (s), and $V_i(\tau)$ is the volume of runoff generated at input cell i and time $\tau$ (m$^3$). It is important to note that the flow path response function $U$

($t$) calculated based on the travel time to the outlet of each sub-catchment from each cell ($t_0$) and its standard deviation ($\sigma_0$) was in the form of first passage time distribution [19].

$$U(t) = \frac{1}{\sqrt{2\pi\sigma_0^2 t^3 / t_0^3}} \exp\left[-\frac{(t - t_0)^2}{\frac{2\sigma_0^2 t}{t_0}}\right] \tag{2}$$

The resulting response function was a three-dimensional (3D) matrix where x and y represented the coordinates of each cell and z represented the lag time. This 3D matrix played the most important role in the outflow calculation by providing information regarding the percentage of runoff generated from different cells as a function of time. The longer the lag time, i.e., the mean travel time to the catchment outlet ($t_{0\_h}$), the more complex the expected 3D matrix was. For this reason, a methodology for limiting the mean travel time to a certain level was proposed. Therefore, a less complex matrix was generated to be solved for the calculation of runoff. Due to the large number (millions) of pixels available in high-resolution data, it was difficult to find to what extent the $t_{0\_h}$ could be limited without losing too much information. Therefore, a statistical technique for organizing large data, typically used in water resources engineering, was used [43]. Using this method, grouping of $t_0$ was done by distributing them over a certain number of groups, called class intervals, with equal width. According to [44], the number of class intervals ($K$) could obtained by Equation (3).

$$K = 1 + 1.33 \log (n), \tag{3}$$

where $n$ is the sample size (i.e., the number of pixels in a sub-catchment) counted automatically by the model. Subsequently, the smallest and highest value of $t_0$ was considered as the lower boundary of the first class ($t_{0, L}$) and the upper boundary of the last class ($t_{0, U}$), respectively. Then, the range was calculated ($t_{0, U} - t_{0, L}$) and divided by "K". Finally, the relative frequency of each class was calculated by dividing the number of pixels in the range of each group by the total number of available pixels. A threshold was then specified to separate the value of $t_0$ in the latter classes to the upper boundary of the former class. This led to fewer complex matrixes for the calculation of surface runoff with the lowest number of changes to the original input data. The process was done automatically in the pre-processing section of the surface runoff calculation in WetSpa-Urban.

### 2.4.4. The Model Set Up

The general scheme for modeling in WetSpa-Urban is represented in Figure 3. Two types of input data were needed, i.e., meteorological data (precipitation and potential evapotranspiration) and spatially-distributed data in the format of ASCII maps for the land-use, soil, DEM, and catchment boundary maps.

In the WetSpa-Urban pre-processing, the tool prepared the land-use, soil, and DEM maps for each sub-catchment based on the catchment boundaries map generated with the new sub-catchment delineation concept, as explained previously. The user had to specify the number of derived sub-catchments, as airborne hyperspectral images have the advantage of combining a high spatial and a high spectral resolution enabling detailed land-cover classification. In this study, a high-resolution airborne prism experiment (APEX) image with 2 m resolution was used to create the land-use map. Satellite-based multi-spectral data exists (Sentinel-2, Landsat 8, etc.), however, because of the loss of spatial and spectral resolution, they are not the preferred choice for urban applications, although they might still be considered a cost-effective alternative. In addition, aerial photography can capture high spatial detail with lower spectral detail compared to other methods, which limits the number of land-cover classes that can be identified. For the elevation map, light detection and ranging (LIDAR) data collected from the Urban Information System Digital Train Model (UrbIS-DTM) was aggregated to 2 m resolution. The same pixel size was used for the soil map. A requirement of the tool was that

the maps had the same coordinate system and pixel size and be in ASCII format. In addition, the new catchment division map based on surface and sub-surface features was needed.

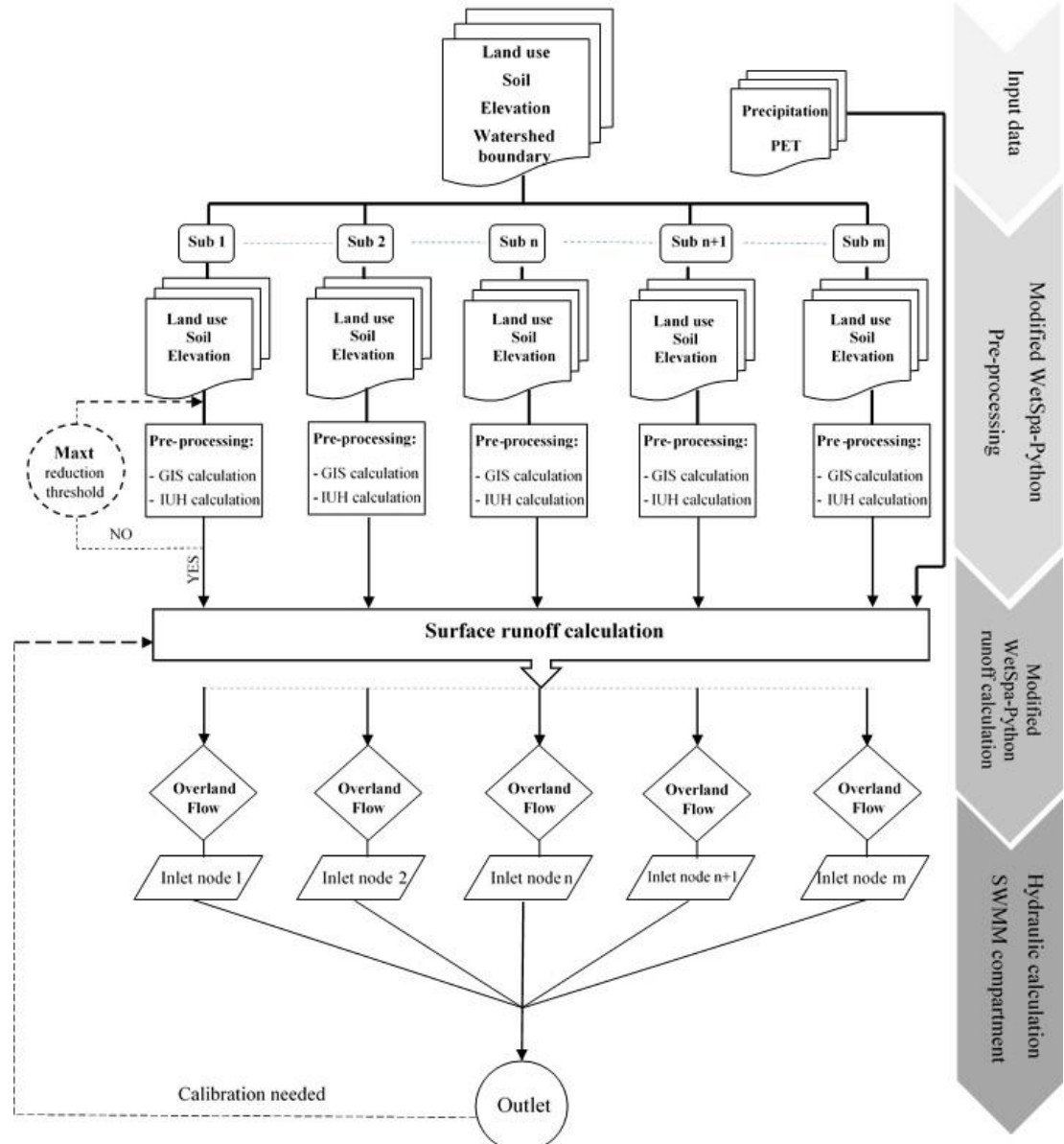

**Figure 3.** The general scheme of the WetSpa-Urban modeling tool.

Next, the grid-based spatially distributed parameter maps, including Manning coefficient, potential runoff coefficient, $t_{0\_h}$, $\sigma_0$, etc., were automatically calculated by the use of the default parameter tables in WetSpa-Python and a series of thresholds in the course of GIS pre-processing stage [17]. During the IUH pre-processing step, the user could decide whether the maximum time of concentration (Maxt) should be reduced to increase the speed of the surface runoff calculation. In case the user decided to reduce Maxt, the reduction threshold [%] had to be defined by the user. This procedure was done manually and for each sub-catchment separately.

In the next stage, the generated parameter maps, together with the grid-based spatially distributed meteorological data and the generated 3D matrix of the response function, were used to calculate the runoff from each sub-catchment. The 10 min reference crop of evapotranspiration was calculated based on the Penman–Monteith equation [45] from climatological records of solar radiation, temperature, humidity, and wind speed measured at the nearby Uccle station. Then, the 10 min measured

precipitation of the only station located inside the study area (Dépôt Communal) and calculated reference crop evapotranspiration of the Uccle station for two different periods in 2015 (January–February and July–August) were used for the calibration period. The remaining months of 2015 were used as a validation period. These two periods were selected for calibration mainly due to the second period (summer) being characterized by high rainfall intensity during convective storms, while the first period (winter) had the lowest number of dry periods together with high average rainfall intensity. For the water balance simulation and calculation of the runoff at the catchment outlet, the 8 global space and time invariant parameters [17,19] were defined for the whole catchment. The full list of these global parameters and their suggested ranges are presented in Table A1, Appendix B [17,46]. In this study, due to having only one flow measurement station at the outlet of Watermaelbeek catchment, the global calibration parameters were identical for whole the catchment area. However, having more flow measurement stations spread over the catchment area led to different parameter values for each sub-catchment draining to different measurement stations, which resulted in more accurate flow estimations and water balance measurements.

Then, Wetspa-Urban automatically prepared all necessary input files per sub-catchment outlet or sewer inlet nodes (where runoff has been routed to) to enable the simulation of water in hydraulic structures by SWMM. As a final step, the surface runoff was routed toward the sewer network by assigning the discharge from each sub-catchment to a specific inlet manhole. By running the hydraulic compartment, the flow at the outlet was measured. In order to improve the efficiency of simulation results, the 8 global parameters (in the surface runoff calculation) and/or conduit's roughness coefficient were modified during the calibration stage.

## 3. Results and Discussions

### 3.1. The Sub-Catchment Delineation

As WetSpa(-Python) was not developed specifically for modeling in urban areas but was designed to estimate the water balance at the catchment scale, the model input was adjusted by combining the DEM and sub-catchment division maps based on the sewer network. The first was used to analyze the flow direction above-ground, and the latter divided the sub-catchments based on the drainage area of each inlet node.

As illustrated in Figure 4B, the red polygon lines represent the boundary of each sub-catchment based on the drainage network corrected by administrative boundaries. In this figure, delineated sub-catchments based on the DEM (shown in different colors) were generated by specifying inlet nodes as an outlet of each sub-catchment. To modify the given boundaries in order to redirect flow to the right outlet node, some sub-catchments were disaggregated and redirected. On the other hand, although some sub-catchment boundaries were administrative, and since it was better to consider them as distinct sub-catchments due to the scarcity of detailed information regarding the drainage network, they could not be considered separately. Consequently, those sub-catchments were aggregated, which helped to reduce computation time as well. For example, sub-catchment number 36 (shown in gray) (Figure 4B) was covered mainly by forested areas in the western part and was aggregated without significantly affecting the stormwater calculation. This was done as all sub-divisions of sub-catchment number 36 had the same characteristics and they were not as important as urban areas in terms of flooding impact. Thus, the four specified sub-catchments by administrative boundaries (shown by the red polygon inside the gray-colored area) were merged into one (sub-catchment number 36).

Sub-catchments 37 and 30 were an example of disaggregation. The total area of sub-catchments 37 and 30 looked to be approximately equal to the one provided by the local authority. However, by comparing changes in their elevation (Figure 4A), the flow direction followed the opposite direction in each of them. On the one hand, relying only on the DEM would have merged sub-catchment 37 into 36, which would have led to an incorrect volume of water routed through the lower sewer branch, thereby causing a double-peak phenomenon at the end of the simulation. On the other hand,

considering sub-catchments 37 and 30 as one sub-catchment (large red polygon) would have caused an error in WetSpa-Urban, as the software only accepts sub-catchments with one outlet. To solve this issue, it was disaggregated into two zones and then the runoff from each was redirected to the correct inlet node. Finally, a new catchment map containing 38 sub-catchments (the colored areas) was generated. Subsequently, the discharges from some sub-catchments were combined and routed through 32 inlet manholes due to the limitation of not having all of the details regarding the sewer network (all inlet nodes).

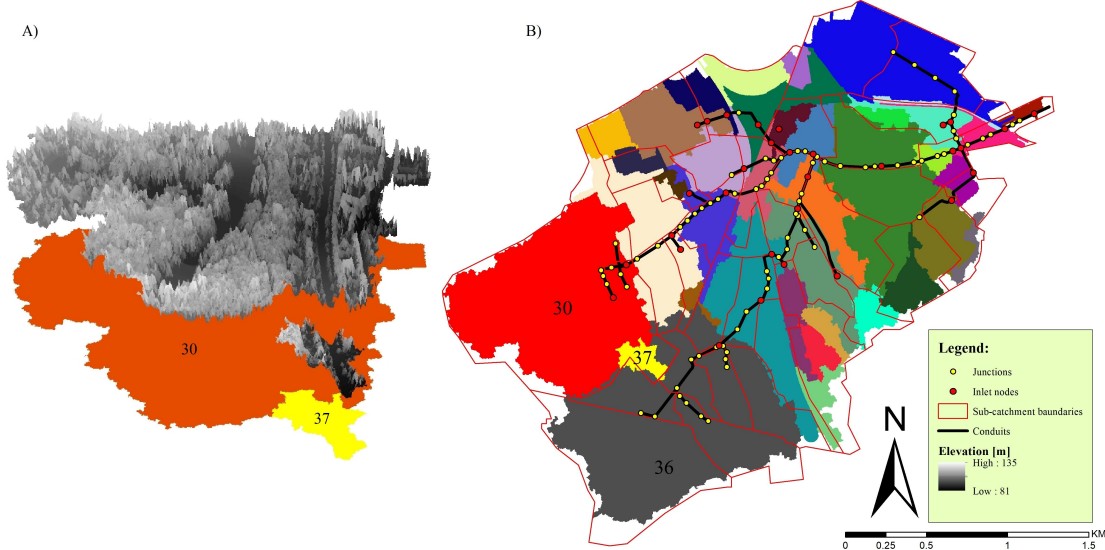

**Figure 4.** Sub-catchment delineation. (**A**) The 3D elevation of sub-catchment 30 and 37; (**B**) the sub-catchment delineation based on the drainage network and DEM.

### 3.2. The Calculation Time Speed-Up

The multithreading approach used in WetSpa-Urban made the surface runoff calculation two times faster compared to WetSpa-Python. As a result, the time needed for calculation of surface runoff decreased from 10 to 4.2 s on average for each time-step. For a simulation of one year with a 10 minute time-step, the calculation reduced from 146 to 61 h (85 h in total), which was achieved by using a personal computer with an Intel Core i7-3610 QM CPU at 2.30 GHz and 12 GB random access memory (RAM).

Secondly, optimizing and rewriting the code was done to overcome the low performance of the surface runoff calculation. By comparing the calculation time of pre-processing with the stormwater calculation component, the pre-processing was much faster than in the latter. Subsequently, profiling the code was done in order to find out which part of the code consumed most of the calculation time. The results showed that the formula to calculate flow for each pixel for different IUHs was very complicated. To avoid very long and repetitive calculations for each pixel within a loop, the calculations of fixed values were moved outside of the specific loop. This simplification and rewriting of the overland and interflow equations increased the model running efficiency time by 30%.

Last but not least, the reduction in the maximum time of concentration "Maxt" was achieved by limiting the mean travel time to the catchment outlet ($t_{0\_h}$) to a certain threshold for sub-catchments with high Maxt. As an example, in sub-catchment 2, Maxt ranged between 0 and 5.25 (per 10 min time-step). Then, the values were divided into 14 groups with a fixed interval of 0.375. After that, the number of pixels within each class was counted, followed by calculating the relative frequency. The results showed that the last six classes (ranging from 3 to 5.25 [per time-step]) contained only 41 out of the total pixels in this sub-catchment. In other words, by reassigning the "$t_{0\_h}$" value of the last 1% pixels in this sub-catchment to an upper limit of the 8th class, which was 3, the Maxt was reduced by a factor of 2. In this catchment, the Maxt varied between 2 and 11 for the different sub-catchments,

but 4 and 5 were the most dominant values. By reassigning the value of pixels with unrealistically high "$t_{0\_h}$" to a certain level (with a normal distribution and 99% confidence interval), shorter IUHs were obtained for sub-catchments with high Maxt values. As can be seen from Table 2, the average Maxt for the whole catchment reduced to 4 with a maximum of 6 for two sub-catchments. In other words, reassigning a lower value to less than 1% of pixels in the catchment led to a decrease of approximately 20% in average total time to the outlet of the sub-catchments. This method considerably improved the model performance without adding large uncertainty, only a very limited effect on lag time.

**Table 2.** Reduction in Maxt and the average total time to the WMB catchment by limiting the mean travel time to the catchment outlet ($t_{0\_h}$).

|  | Original | Reduced |
|---|---|---|
| Number of Maxt (number of time-steps) | 5 | 4 |
| Avg. total time to sub-catchment outlet (min) | 50 | 39 |
| Unchanged pixels (-) | 1,533,786 | 1,531,524 (−0.14%) |

*3.3. The Stormwater Calculation*

The first step in modeling with WetSpa-Urban was to divide the soil, DEM, and land-use maps into smaller segments (sub-catchment level) based on the given catchment boundary map. Next, the generated maps for each sub-catchment were used for the pre-processing section of the surface runoff calculation by providing a wide range of spatially distributed parameter maps. Therefore, the user had to first define the series of input thresholds for each sub-catchment in order to run the sub-catchments in parallel and check whether there was an error in generating the flow direction map for any of the sub-catchments. In case of an error in generating the parameter maps, the model stopped working and there was a need to refine the parameters for each erroneous sub-catchment separately. The errors mainly occurred from differences in the spatial extent of the sub-catchments. To solve this, the sinks were filled by changing their respective thresholds to avoid errors in generation of the flow direction and accumulation maps, which are the backbone of all other parameter map calculations.

The stream net parameter map had a significant effect on the peak's lag time and concentration time of each sub-catchment. In urban areas, the stream net acted as an open channel with predefined Manning roughness coefficients in order to route the runoff to the outlet of each sub-catchment. To get a more realistic shape of a stream network, the value of "stream net threshold" in each sub-catchment was calibrated by visual interpretation. The larger the "stream net threshold", the lower the expected number of open channels. Figure 5A represents an unrealistic shape of a stream net and Figure 5B shows the more realistic stream pattern by assigning threshold values of 2 and 200, respectively. Using the stream net map presented in Figure 5A in the surface runoff calculation, conduits over the roof pixels and in back yards of each parcel were observed, which made it unrealistic. This caused faster runoff and peaks tended to appear earlier in the simulation results compared to observations. By calibrating the stream net threshold, the maps illustrated in Figure 5B were derived. In this map, the channels were a good representation for available conduits in the streets, as they only followed the street pattern.

The precipitation and potential evapotranspiration data (2015) together with the generated physically-based parameter maps were used to run the runoff calculation module. To run this module, eight global parameters (listed in Table A1, Appendix B) were defined. Before running the hydraulic calculation (SWMM compartment), the simulated overland flows from each sub-catchment were diverted to its relevant inlet junction. In this study, the outflows from 38 sub-catchments were routed through 32 inlet nodes. As an example, the flows from sub-catchments 31, 35, and 38 were aggregated at inlet node 26. The same happened with sub-catchments 1 and 12, which were routed through inlet node 1.

The whole procedure was repeated with different values for the eight global parameters to calibrate and validate the model. In this case, Krun and Pmax were found as the most sensitive parameters in the process of manual calibration, and the impact of the changes in these parameters

on the results can be found in [17]. However, to have a better understanding of the impact of each parameter, an in-depth sensitivity analysis should be performed, which is beyond the scope of this study. The calibrated and validated global parameters are presented in Table 3. These parameters were used only for this catchment with the specified resolution (2 × 2 m). The flow simulation versus observations for a selected period in January 2015 can be seen in Figure 6.

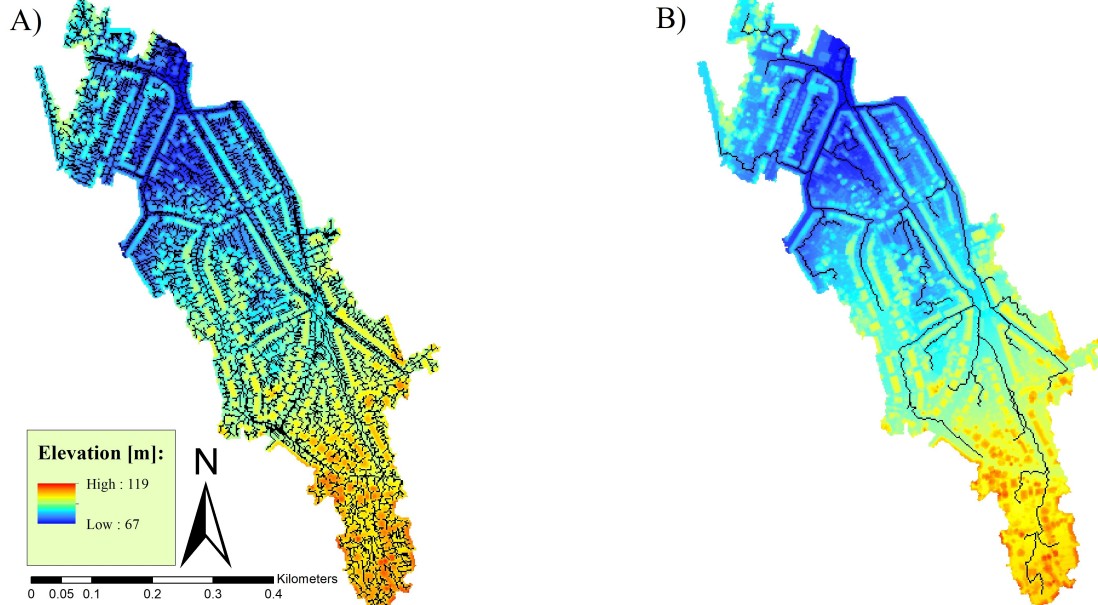

**Figure 5.** Stream net parameter map. (**A**) Unrealistic shape of a stream net before calibration; (**B**) realistic shape of a stream after calibration.

The low flows, mainly the dry weather flow (DWF), were simulated well, but for the peaks with high flows, some overestimations and underestimations were observed. It is also worth mentioning that the timing of most of the simulated peaks fit quite well with the observations, therefore, the parameter maps, especially stream net and the $t_{0\_h}$ maps, were generated accurately. The Nash–Sutcliffe efficiency (NSE) [47] was calculated for both the calibration and validation period and were 0.88 and 0.85 respectively, showing good results.

Figure 7 illustrates the correlation between simulated flow and independent peaks with observations. As can be seen from Figure 7A, the simulated versus observed outflow from the Watermaelbeek catchment presented a significant positive correlation ($R^2 = 0.9$, equation: y = 1.031x + 0.003 (the numbers in the equation presented in the figure were rounded to one decimal place)). Moreover, there was a correlation between the time of errors and the flow values. To perform a statistical investigation independent of time, the time series was split into the specific number of independent events and the highest value was selected as the peak flow. The extracted peaks from the observations and the simulation were plotted using box-cox transformation in Figure 7B [48,49]. By applying the box-cox transformation to the observed and simulated values, the standard deviation of the errors became independent of the discharge magnitude. In this manner, all low and high discharges had the same standard deviation, therefore, the dotted lines were drawn parallel to the bisector. For more details on the box-cox transformation and the method of its calculation, please refer to the water engineering time series processing tool (WETSPRO) user manual [50] and/or [48,49]. As shown in Figure 7B, the bias in the model is represented by the distance between the bold line and bisector, which had a positive bias in this case. The dotted lines represent the one-time standard deviation, which reflected the random uncertainty. The homoscedasticity property of this graph represents the independence of the magnitude of the error and model output. The results showed that the highest peak was underestimated, however, the rest were simulated well.

**Table 3.** The global calibration parameters of the WetSpa-Urban modeling tool for the WMB catchment test case.

| Parameter | Symbol | Parameter Value |
|---|---|---|
| Correction factor for PET | $K_{ep}$ [-] | 0.75 |
| Actual runoff coefficient correction factor | $K_{run}$ [-] | 24.67 |
| Rainfall intensity scaling factor | $P_{max}$ [mm h$^{-1}$] | 10 |
| Initial soil moisture correction factor | $K_{ss}$ [-] | −0.43398 |
| Interflow scaling factor | $K_i$ [-] | 0.83 |
| Initial groundwater storage parameter | $G_0$ [mm] | 1000 |
| Base-flow recession coefficient | $K_g$ [h$^{-1}$] | 0.000053 |
| Groundwater storage scaling factor | $G_{max}$ [mm] | 1050 |

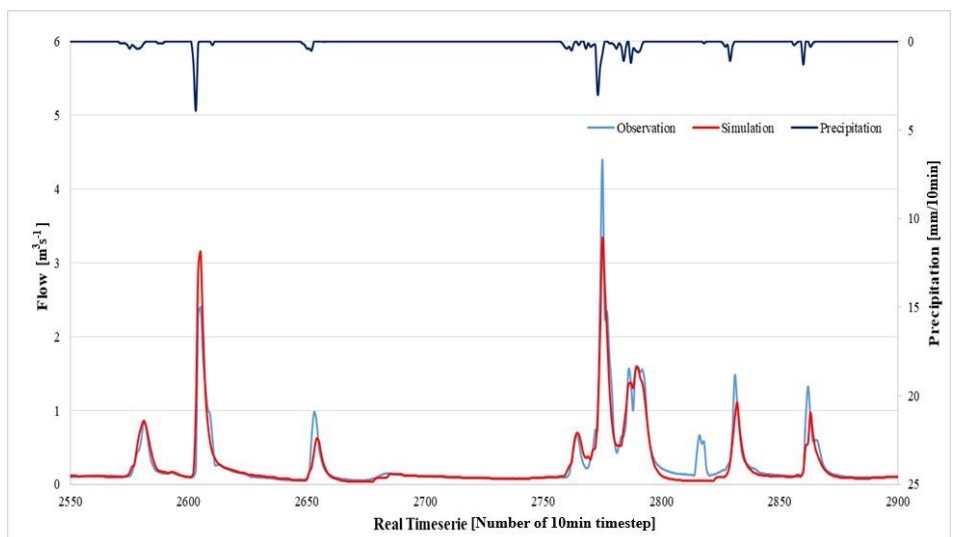

**Figure 6.** Simulation by WetSpa-Urban versus observation at the outlet of the WMB catchment.

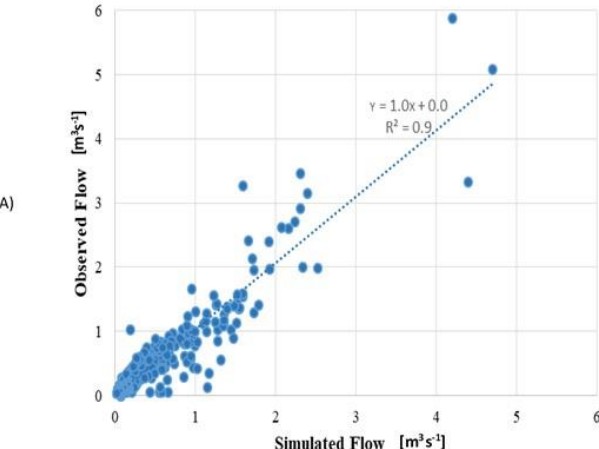

**Figure 7.** *Cont.*

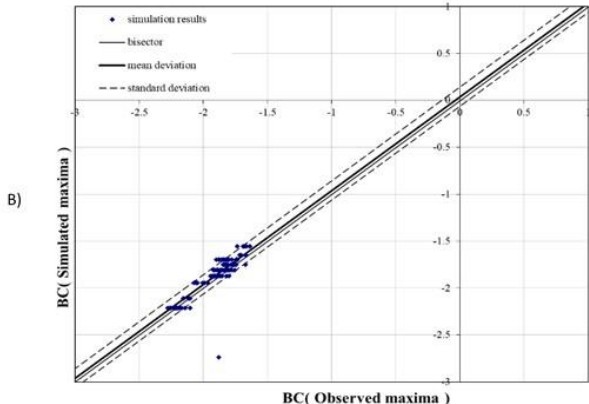

**Figure 7.** Statistics of the simulation versus observation flow at the outlet of the WMB catchment. (**A**) Scatter plot; (**B**) box-cox transformation of observed versus simulated peaks.

## 4. Conclusions

In this study, the adapted version of WetSpa-Python suitable for urban catchments, which is called WetSpa-Urban, was introduced. The main aim of WetSpa-Urban is to increase the predictive power of runoff simulations in urban catchments. This software couples two existing tools: WetSpa-Python and SWMM, along with some modifications. The core code of WetSpa-Python was used for the runoff calculation and the sewer processes were modeled using SWMM.

As urban areas have a high level of heterogeneity, the runoff calculation was done in a spatially-distributed method and the surface runoff was calculated at the outlet of each sub-catchment, separately. The sub-catchment division was not restricted to the slope of land but also considered the slope of the sewer network, even if it did not follow the surface elevation and slope. Therefore, modifications of the provided boundaries based on the drainage network division with the DEM were performed to ensure the correct volume of surface runoff was routed through each branch of the sewer network. A more accurate routing of the stormwater in the sewer system avoided unrealistic magnitudes and timing of peaks at the outlet of the whole catchment, as observed with the WetSpa-Python simulator, which was originally developed for river catchment areas in general.

Due to the high level of input details and low performance of Python compared to the other programming languages, relatively slow calculation speeds for continuous modeling were found. Therefore, a model speed-up was performed in three steps, i.e., a multithreading approach, optimization of codes, and a reduction of "Maxt". The multithreading technique allowed the calculation of the flow at each pixel for different IUHs in parallel, leading to a 60% reduction in the running time of the model. Simplification and rewriting of the equations and codes were done to optimize the script, thereby giving a 30% faster calculation time. Lastly, the reduction of the mean travel time of pixels to the outlet led to a lower "Maxt", resulting in a less complex matrix and faster calculation speeds. After modification of $t_{0\_h}$ for 2262 pixels with a value of more than 6, the total time to the catchment outlet reduced by 11 minutes (20%). In other words, by reducing a high $t_{0\_h}$ value in 13 sub-catchments with a value more than 6 (per time-step) (average $t_{0\_h}$ in the whole catchment), all input data was kept the same as the original, with changes in $t_{0\_h}$ of less than 1% of the total pixels. Compared to other methods, in which the sinks or very deep pixels are selected manually and then filled to reduce abnormal "Maxt", in the new method, no artificial changes were added to parameter maps generated from DEM, which may result in uncertainty to the model output. As a result, only two sub-catchments had a maximum time of concentration of 6 (over the average). Overall, the WetSpa-Urban was 130% faster in calculation speed compared to the WetSpa-Python, which makes it ideal for both design and continuous modeling in urban areas. Although, using the original version of WetSpa written in FORTRAN could be a choice for coupling to avoid performing these changes for speeding up the calculation time of the modeling, this leads to loss of model flexibility as the Python version has a process-based model structure and can handle a wide range of spatial and temporal input data.

After all of the modifications, the modeling was done in three separate stages: pre-processing, surface runoff calculation, and routing through hydraulic structures. In the pre-processing section, 33 grid-based parameter maps were generated, and later they were used in the runoff calculation together with eight global parameters. Then, the flow from each sub-catchment was routed through the desired inlet junctions. The results showed good simulation quality of the model, with an NSE of 0.86. Compared to the research done by [18] for the WMB catchment at an hourly time-step with the original WetSpa, a significant improvement in the simulation of flow was observed (NSE = 0.70). Although the model performed well, with a bias of 0.06, a slight over-estimation was observed in the simulation results. A sensitivity analysis could be performed, which would allow for reduced computation time of an (semi-)automatic calibration, such as model-independent parameter estimation (PEST) [51], which could improve model performance. These findings for WMB catchment confirmed that WetSpa-Urban provided a realistic representation of the discharge dynamics for urban areas even for a catchment without detailed information regarding sewer systems and sub-catchment representation. Although WetSpa-Python and SWMM are applicable to various conditions globally, the software described in this paper should be tested in other urban catchments with different characteristics before concluding that the tool is widely applicable.

Although the scarcity of data regarding the connectivity of sub-catchments and inlet nodes together with detailed information regarding sewer networks is a problem, in this study, the newly developed tool performed better in comparison to WetSpa-Python. This shows the applicability of WetSpa-Urban in runoff simulations in study areas with limited data. However, as a result of lost storage volumes due to missed pipes, modeling of floods and CSO may not be as accurate. In other words, such modeling with major sewer networks could be used for lumped hydrological (long-term, continuous) modeling; however, this would not be suitable for detailed evaluations of sewer network performance. Therefore, as WetSpa-Urban can handle sewer networks and sub-catchments at any level of detail, it is advisable to use this model with a higher number of sub-catchments and more detailed sewer networks to obtain better results.

It is concluded that the level of detail regarding hydrological processes modeled by WetSpa-Urban is higher compared to using WetSpa-Python or SWMM individually. In general, WetSpa-Urban tries to overcome the shortcomings of conceptual stormwater calculations and rough estimations of ET in SWMM. Moreover, having multiple outlets (at the sub-catchment level) and considering drainage networks in its calculations make WetSpa-Urban more suitable for urban studies in comparison to WetSpa-Python.

Additionally, having a graphical user interface (GUI) is another added value of the newly developed software adapted for urban areas. In general, WetSpa-Urban was developed in a way to be applicable in urban studies for design purposes and long-time series without any limitations. It can be used with high or coarse resolution input data and the number of sub-catchments can be varied based on user selection. The other advantage of this tool is the possibility for further development to add a module for implementation and modeling of low impact development (LID) practices due to its free availability and the usage of remotely sensed data for runoff calculation. This module, which is called the LID locator tool [34], was developed in order to find the best locations for implementation of the most cost-optimized combination of four types of LIDs. This could be an important accomplishment for urban planners in order to improve the social and environmental aspects of urban land by further developing this software.

## 5. Recommendations

As explained in this article, in the WetSpa-Urban modeling tool, the interaction between the sewer networks and groundwater was not considered, which may be a serious issue in cases with high groundwater tables (exfiltration) and/or old sewer networks (infiltration). Due to a recent tendency to introduce LID practices to cope with the problems associated with climate change and urbanization,

thereby resulting in more permeable surfaces in urban areas, adding groundwater and sewer interaction processes to the next version of this tool is recommended.

Great progress in speeding up the calculation time was achieved by the three methods used in this study, but other methods could be applied to achieve the same objective. One option could be to convert the WetSpa-Urban codes to a faster programming language, such as C, C++, or FORTRAN, compared to the current Python version while maintaining the process-based model structure. Another option could be to incorporate machine-learning and computational intelligence methods to better predict the flow in a faster manner, thereby making it suitable for real-time continuous modeling [52–54].

Although using different formats of input data did not have any impact on the quality of our results, to make WetSpa-Urban more user-friendly, adapting the tool to be capable of analyzing and handling different formats of input data is recommended. Finally, it is advisable to incorporate a reliable automatic calibration method into WetSpa-Urban in order to increase its performance.

**Supplementary Materials:** The following are available online at https://github.com/VUB-HYDR/WetSpa-Urban. The WetSpa-Urban scripts, as well as 2 m resolution land-use, soil and DEM maps and 3 months meteorological data are available as sample data on GitHub (https://github.com/VUB-HYDR/WetSpa-Urban). Descriptions of the WetSpa-Python installation needed for running the WetSpa-Urban modeling tool and all preinstalled software packages this model requires are also provided within the GitHub page.

**Author Contributions:** Conceptualization, N.R.H., B.V., and W.B.; methodology, N.R.H.; software, N.R.H., C.W.; validation, N.R.H.; writing—original draft preparation, N.R.H.; writing—review and editing, I.W., W.B., B.V., A.v.G., and C.W.; supervision, B.V., A.v.G., and W.B.

**Funding:** This research received no external funding.

**Acknowledgments:** The meteorological dataset was provided by the Royal Meteorological Institute (Uccle) and the Brussels Company for Water Management (SBGE-BMWB) via Flowbru.be. We thank Ing. Guido Petrucci for assistance in providing details about the sewer network and the existing SWMM model. The basic information on the sewer network was provided by VIVAQUA, and the network was refined based on available communal paper maps. The high-resolution DEM was derived from the freely available Urbis-DTM product in GRID format (https://bric.brussels/en/our-solutions/urbis-solutions/urbis-data/urbis-dtm?set_language=en). The distributed soil texture map was a digital version of the analogue soil map of Belgium, produced by AGIV and freely available at DOV (https://www.dov.vlaanderen.be/geonetwork/srv/dut/metadata.show?uuid=5c129f2d-4498-4bc3-8860-01cb2d513f8f).

**Conflicts of Interest:** The authors declare no conflict of interest.

## Appendix A

The parameter maps were divided into land-use, soil, and topography together with thiessen polygons, flow routing parameters, the potential runoff coefficient, and depression storage capacity.

- Land-use based parameter maps: root depth, interception capacity, and Manning coefficient
- Soil based parameter maps: conductivity, porosity, field capacity, residual moisture, pore distribution index, wilting point, and initial soil moisture
- Topography based parameters: mask, flow direction, flow accumulation, stream network, stream order, slope, hydraulic radius, stream links, and sub-watersheds
- Thiessen polygons: precipitation and potential evpotranspiration
- Flow routing parameters: velocity, $t_{0\_h}$ (mean travel time to the catchment outlet), Delta_h (standard deviation of flow travel time to the catchment outlet), T0_s (mean travel time to the sub-catchment outlet), and Delta_s (standard deviation of flow travel time to the sub-catchment outlet)
- Potential runoff coefficient and depression storage capacity: runoff coefficient and depression storage capacity
- The parameter map's calculation flowchart is presented in Figure A1.

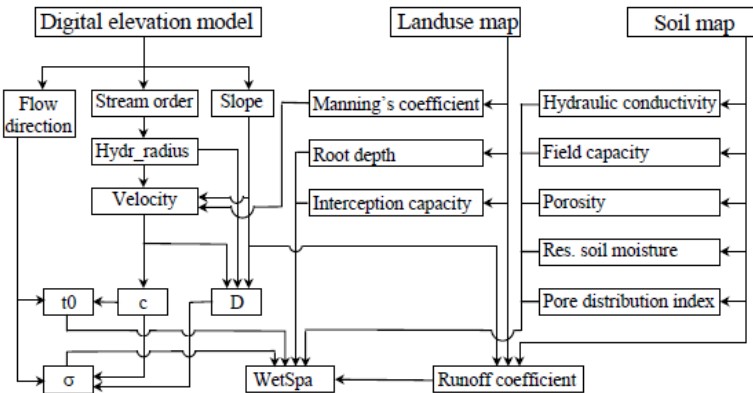

**Figure A1.** Flowchart of calculation of some important parameter maps in GIS pre-processing section of WetSpa-Python [55].

## Appendix B

**Table A1.** WetSpa-Python global calibration parameters and their suggested ranges provided by Berezowski et al. [46] and modified by Salvadore [17].

| Parameter | Symbol and Unit | Parameter Range |
|---|---|---|
| Correction factor for PET | $K_{ep}$ (-) | 0.3–2.0 |
| Actual runoff coefficient correction factor | $K_{run}$ (-) | 0.01–15.0 |
| Rainfall intensity scaling factor | $P_{max}$ (mm h$^{-1}$) | 1–700 |
| Initial soil moisture correction factor | $K_{ss}$ (-) | Negative or 0.1–3.0 |
| Interflow scaling factor | $K_i$ (-) | 0.1–20.0 |
| Initial groundwater storage parameter | $G_0$ (mm) | 1–500 |
| Base flow recession coefficient | $K_g$ (h$^{-1}$) | $1.10^{-6}$–$1.10^{-1}$ |
| Groundwater storage scaling factor | $G_{max}$ (mm) | No specific value |

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
