# Peer review of "WetSpa-Urban: An Adapted Version of WetSpa-Python, A Suitable Tool for Detailed Runoff Calculation in Urban Areas"

_water, doi:10.3390/w11122460_

Round 1

Reviewer 1 Report

This paper links the WetSpa-Python runoff modelling tool to SWMM. It's main contributions, as stated by the authors, are:
1. Link the WetSpa tool for surface runoff calculations to SWMM to allow for the simulation of overflow in a detailed manner
2. Introducing a new approach for sub-catchment delineation by combining surface flow paths and drainage catchments linked to sewer pipes
3. Optimize computational speed of the package.

Regarding these 3 statements, my conclusions after reading the paper, are the following:
1. It is not clear what knowledge gap the authors actually aim to adress. Existing software packages do allow for detailed simulations of surface flows. We can simulate runoff, infiltration and evaporation processes for different area types. This is true for the SWMM simulator which is also used by the authors, and even more so for commercial software packages such as InfoWorks or MIKE Urban. The latter also allow for the consideration of multiple subcatchments linked to the same node, as well as the simulation of runoff processes on 2D raster surfaces, as well as flexible commbinations between these approaches. Thus, I cannot identify a novel contribution of the work presented here.
2. From the paper, it is difficult to understand how exactly catchments are delineated and how exactly the delineated catchments are considered in the runoff simulation. My understanding is that the authors use catchments delineated by the municipality / utility company as starting point and then subdivide these according to the surface flow paths. If that is correct, then the approach is conceptually wrong, as it does not consider the fact that the link between impervious areas and sewer inlets does not follow surface slopes, but is created through underground pipes.
3. The first two approaches for improving computational speed are simply programming details that are of limited interest to the reader. For the grouping approach, no justification is provided for why the subdivision into K intervals would be a valid approach. The authors document simulation times in the order of several days for continuous simulation of a one year period. This is several orders of magnitude slower than state-of-the-art simulation packages such as MOUSE or KOSIM.

In my opinion, the paper requires substantial revisions to clarify how catchment delineation approaches and runoff simulation are implemented exactly. Even if an improved description is provided, I am not sure if a novel contribution can be identified. I therefore suggest rejecting the paper.

-----------------------
Some detailed comments:
Introduction:
-there is some mix up in the terminology - rainfall runoff models are used to simulate the amount of runoff generated, but the authors sometimes seem to use the term synonymously for the whole simulation package (including 1D sewer pipe model)
-the introduction is very lengthy and cites various works / reviews, but without clarifying what existing gap in rainfall-runoff modelling the presented work addresses

Methods
-l 156: what is the point of performing runoff simulations in high spatial detail if only a trunk sewer system is considered?
-Sect. 2.4.1 - it is not clear what data is used exactly for catchment delineation in which steps, and how the delineated catchments subsequently feed into runoff calculations
-Sect. 2.4.2 - Fig. 4 and the first part of this section can be removed. From line 286, it is not clear at what spatial resolution the runoff calculations are performed, how exactly the subdivision into k intervals is performed, and why this is a valid approach.

Results
-Fig. 6 - why are only some of the nodes used as inlets? In general, I have difficulty understanding which catchments are used for simulation. Is it the red polygons? Is it the coloured areas? Are both used, but in different parts of the simulation? Is the runoff simulation performed on a 2D grid - but then why delineate catchments?
-l 454: it is not clear what parameter the threshold values refer to
-l 457: how is the calibration performed?
-l 490 following: the box cox transform is not documented. It is also not clear how it introduces an error standard deviation that is independent of the discharge. I'm also not sure why it would be important to know the residual standard deviation in this context.

Code availability
-Since WetSpa-Urban/Python is an open source tool, it certainly is available online? The developed code should be referred to from the paper?

Author Response

Dear reviewer one,

We thank you very much for your constructive feedback. We hereby answer to your comments using the line and figure numbers of the revised manuscript with track changes. Additionally, we replied to the commented pdf in the attached file.

Best regards, the authors.

Reviewer 2 Report

Formal errors, recommendations and comments:

Very well written from formal point of view, just few reccoemndations:

Figures 3 - 5, 8, 9, as well as in eq. 1, 2: if possible do not use raster images. The numbers and text are blurred and sometimes difficult to read. 8: use standard notation “m3.s-1 “instead of “CMS” Units of the horizontal axis –it is the time, or the number of 10 min. timesteps? Pls. be more specific 3: even if it's just a sketch, the sewer pipe seems to be drawn on the ground level and just above the river, it looks very unprofessional 9A – the equation in this Fig. slightly differs from the eq. in text (line 486) Line 486 – R2 – pls use superscript R2

Logical errors, recommendations and comments:

Is there in the described model considered the interaction between the groundwater and the sewer system (exfiltration, infiltration)? According Fig. 3 is such interaction not assumed, however similar process e.g. the groundwater flow into river is assumed. Especially the infiltration into the sewer system can be a serious issue. Typically, the infiltration is included in the wastewater production or as a constant value, but the amount of the infiltration water in long – term modelling is dependent on the groundwater level. In case of pervious ground and ground water level increase (e.g. due to the large infiltration) will also increase the infiltration flow into the sewerage system. I believe that it is not so much necessary to emphasize the acceleration of calculations and the reduction of calculation time by using more effective code compiler (chapter 2.4.2, Fig. 4), but on the other hand the mention of the reduction of the calculation time (lines 285 - 314) is correct and shall remain in the paper. Line 156: “The sewer network is simplified and only the main pipes are modelled in this study.” Please add into the text the lengths and volumes, eventually more basic information about the real and simplified network. The authors shall discuss the consequences and errors, arose by such simplification. My experience shows that such approach can be quite precise regarding the modelled flows due to precise, but in fact inaccurate model parameter calibration. But in cases of modelling floods, surcharge flows and CSO events (CSO events and volume) I have serious doubts about accuracy because of missing storage volume due to the sewer system simplification. I have similar doubts regarding the inlet nodes (Fig. 2), where the distance between the nodes is about 500 m – does it mean, that the surface runoff is about 250 m? This is not realistic and I think, that there will be similar situation like in previous point - precise, but in fact inaccurate model parameter calibration (runoff concentration time, travel time, surface runoff velocity etc.). The authors should more emphasize the fact, that such modelling can be used for lumped hydrological (long-term, continuous) modelling, but it is not suitable for task mentioned in previous point – detailed proof (check) of the sewer network performance (hydraulic performance check, flooding, CSO etc.) Last Chapter – Discussion contains rather conclusions than discussion. Discussion topics are partially spread over in chapter 3 – Results. Not sure, if the model is available for the public (free download) – if yes, such information can be very interesting for the readers, eventually check this issue with the associate editor.

Author Response

Dear reviewer two,

We thank you very much for your constructive feedback. We hereby answer to your comments using the line and figure numbers of the revised manuscript with track changes. Additionally, we replied to the commented pdf in the attached file.

Best regards, the authors.

Reviewer 3 Report

Review

WetSpa-Urban: an adapted version of WetSpa-Python, a suitable tool for detailed runoff calculation in urban areas

General comments:

Nahad Rezazadeh Helmi et al. describe an extended version of the WetSpa-Python hydrological for urban areas, the WetSpa-Urban model. It combines the strength of both the WetSpa model for hydrological processes at the land surface and EPA SWMM, which is best suitable to describe non-stationary flow in drainage networks. The added value of the coupled model and its application in case study in Belgium are described very well by the authors. Through an increase in model efficiency, the authors prove that the model extension WetSpa-Urban is more suitable in urban areas than the original Wet-Spa-Python model. The development of WetSpa-Urban is a nice piece of work, which fills a gap in my opinion. I could imagine that it is of great value for the community! Moreover, the paper fits well into the scope of Water and I believe it is interesting for the readers of Water. However, in my opinion, the paper needs some revisions, before it is ready for publication. Please find my comments below:

The paper could be more concise: For instance, Figure 1 is not really needed to understand the paper. There is a mixture of methods and results that must be avoided in my opinion. The results section includes a summary of parameters that have never been introduced before. I agree that this might be part of the results. However, here I would expect some information about the model performance (which then follows) rather than an extensive description on how sub-catchments have been merged. Added value. I was wondering if it is possible to include the results of the WetSpa-Python approach in Figure 8? You only refer to an earlier study. Since the authors of the cited study are co-authoring this manuscript, you could consider including some more details of the earlier results. This would be a major improvement of your paper. You should provide more information on rainfall and Evapotranspiration (ET) data (see comments below). Open science. You state that your model is freely available, which is great. I would expect at least an availability section which describes how readers can get the model.

Specific comments:

L41p: Why has urbanization an impact on slope and soils? At first sight, there is an impact on land-use. It is clear that there are implications on slope and soils too. Are you aware of studies that prove this statement? Please be more specific!

L54p: I would suggest adding the level of detail regarding hydrological processes as well.

L65p: What is ‘medium’ spatial resolution? Please be more specific!

L95p: Is Figure 1 really needed here? Please consider dropping it, since it’s available elsewhere.

L102pp: Please consider aspects other than sub-surface too! Slight modifications of the surface are also relevant in urban areas! Moreover, sub-surface is misleading, since this term is often used for groundwater flow.

L114pp: Is this really an open scientific discussion? Please prove this statement by referring to relevant studies!

L152pp: What is the level of aggregation associated to the average temperatures. Are these seasonal averages? Please be more specific!

L171: What means instantaneous here?

L181: Do you mean that WetSpa is not suitable for long-term simulations? Why then doing all the work here?

L195: I am not sure whether ‘conveyance section’ is correct in this context.

L205: In L226 you add a sub section to section 2.4. Maybe you could introduce 2.4.1 “General description” here?

L206: A ‘precise estimation’ is somewhat contradicting.

L245pp: Please better motivate the utilization of administrative boundaries by adding some more background from a practical perspective. I think this description could be improved here!

L331: Why do we have 33 grid-based parameters in total? I would expect some description about this number in an earlier section!

L340pp: What is the spatial resolution of rainfall data? Is this a single station? This has to be discussed later in the discussion section, since highly resolved rainfall is always subjected to uncertainties! Why do we need ET time series? I would expect that WetSpa is capable of calculating ET! The ET calculation in SWMM is known to be very simplified. Hence, here I would expect a major contribution from WetSpa in the coupled model approach! Please include some more details!

L348p: Here, global parameters are described. These should be described more detailed in the model description section earlier! Are they assigned for each gauging station? How is ponding handled in WetSpa? Is there any consideration of overflow along the road, similar to SWMM?

L370pp: Table 1 should be part of the methods section!

L375pp: I found this description of sub-catchments really confusing (e.g., where is sub-catchment #5, as stated in L384?)! I expected a description of the results obtained utilizing the model.

L385pp: The overlay of layers in Figure 6(a) is confusing in my opinion. Please consider another way of displaying the sub-catchment ids and the DTM!

L390: What are ‘trends of change’? Please rephrase!

L404pp: The computation time is rather slow. Could you please provide some information on the CPU?

L432pp: Why is the reduced MaxT a floating-point number? Does the model account for sub-time steps?

L434pp: I don’t understand the discussion on errors in sub-catchments… What do you mean by ‘errors’?

L448: I agree that this threshold is important. Is this discussion really need here? I think that the readers are aware of the relevance of this parameter.

L462: What do you mean by ‘underground compartment’? Is it the drainage network in SWMM?

L467pp: The parameters should be introduced earlier!

L473pp: The rainfall input could be displayed as bar chart. It’s not mm but mm/time step! Is it the areal precipitation intensity (see my comments on the rainfall data)?

L479: I would suggest providing NSE as a fraction, e.g., 0.86 instead of 86%. Providing a percentage is rather uncommon in case of NSE.

L480: What do you mean by model confidence? Is there any reference other than grey literature, like e.g., a PhD thesis? Your model would be the first model that is perfect :-)

L505p: Why does the model run at the sub-catchment scale? This would be similar to SWMM. Your model is distributed in space. I think you should highlight this in order to better motivate the benefit of your study!

Technical comments:

L29p: speed-up?

L275: speed

L500: study instead of research?

L518: Is it really an increase? Here, I would expect decreasing computation time.

Even though I have a lot of comments, it’s only to help improving the paper. I am looking forward to reading your final published paper!

Author Response

Dear reviewer three,

We thank you very much for your constructive feedback. We hereby answer to your comments using the line and figure numbers of the revised manuscript with track changes. Additionally, we replied to the commented pdf in the attached file.

Best regards, the authors.

Round 2

Reviewer 3 Report

I acknowledge the revisions suggested by the authors. In my opinion the manuscript is now significantly improved! I think that there a few things left which could be considered prior to publication.

L99: lead instead of leads

L101: Please consider rewriting evapotranspiration (like in L104)

L105: Please consider replacing “it” by “these processes”

L108p: I think that this statement is not correct. First, SWMM does not “measure” ET. Second, I am not sure whether ET is only withdrawn from open waters. Here, I would recommend rephrasing the statement on the complexity of SWMM regarding ET calculations. You could claim that SWMM is less complex in terms of ET calculations, compared to WetSpa-Urban, at least, since ET calculations only rely on temperature and no land-use dependency is introduced by SWMM.

L309: speed-up

L375p: Again, what type of ET is needed to run WetSpa-Urban? Is it the crop reference ET, as recommended by Allen et el. (1998)? I think that this question is relevant, since you introduce a (land-use dependent?) PET correction parameter (L208) and the fact that PET might involve meteorological quantities other than temperature only might suggest an increased model complexity. Please add some details on the requirements in terms of ET input data. Your response to my earlier question is fine, but this strength of WetSpa-Urban (compared to SWMM) could be at better motivated here.

References

Allen, R.G., Pereira, L.S., Raes, D., Smith, M., 1998. Crop evapotranspiration - Guidelines for computing crop water requirements (No. 56), FAO Irrigation and drainage paper. FAO - Food and Agriculture Organization of the United Nations, Rome.

Author Response

Dear reviewer three,

We thank you very much for your constructive feedback. We hereby correct the manuscript based on all your constructive comments provided in the second round of revision using the line numbers of the revised manuscript with track changes. Additionally, we replied to the comments in the attached file.

Best regards, the authors.
